# Corrosion Resistance of Aluminum against Acid Activation: Impact of Benzothiazole-Substituted Gallium Phthalocyanine

**DOI:** 10.3390/molecules24010207

**Published:** 2019-01-08

**Authors:** Nnaemeka Nnaji, Njemuwa Nwaji, John Mack, Tebello Nyokong

**Affiliations:** Centre for Nanotechnology Innovation, Department of Chemistry, Rhodes University, Grahamstown 6140, South Africa; joemeks4u@yahoo.com (N.N.); njemuwa2005@yahoo.com (N.N.); j.mack@ru.ac.za (J.M.)

**Keywords:** corrosion inhibition, cyclic voltammetry, open circuit potential, potentiodynamic polarization

## Abstract

This study describes the adsorption behavior of organic inhibitors at the aluminum-HCl solution interface and their corrosion inhibition performance. The organic inhibitors employed are: 4-(benzo [d]thiazol-2ylthio)phthalonitrile (BTThio) and tetrakis[(benzo[d]thiazol-2-yl-thio)phthalo- cyaninato]gallium(III) chloride (ClGaBTThioPc). The corrosion behavior of these inhibitors is investigated using electrochemical and computational techniques. Open circuit potential results reveal predominant cathodic character for the mechanism of aluminum corrosion inhibition by the inhibitors. Inhibition efficiency values from potentiodynamic polarization measurements increase from 46.9 to 70.8% for BTThio and 59.7 to 81.0% for ClGaBTThioPc within the concentration range of 2 to 10 μM. Scanning electron microscopy (SEM) measurements reveal protection of the metal surface from acid attack, in the presence of the inhibitors and energy dispersive X-ray (EDX) measurements show that the most probable way by which the inhibitors protect the metal surface would be by shielding it from the corrosion attacks of Cl^−^ from the acid. Quantum chemical parameters corroborate well with experimental findings.

## 1. Introduction

Aluminum is widely used as a material in automobiles, aviation, household appliances, containers, and electronic devices [1,2]. The resistance of aluminum against corrosion in aqueous media can be attributed to the rapid formation of oxide films on the surface. However, aluminum gets easily corroded in the presence of corrosive acids [1,2]. Studies of the corrosion behavior of aluminum in different aggressive environments have continued to attract attention because of its important applications. Hydrochloric acid is one of the most widely used agents in the industrial sector and it corrodes metals such as aluminum. As such, there is a need to use inhibitors for retardation of the metal dissolution process [3].

Among several techniques used in mitigating corrosion problems, the use of chemical inhibitors remains the most cost-effective and practical method [4]. The development of aluminum corrosion inhibitors based on organic compounds is of growing interest in the field of corrosion chemistry [5]. The reason for this is that even though inorganic substances like phosphates, chromates, dichromates and arsenates, were found to be effective as metal corrosion inhibitors, the major disadvantage is their toxicity and as such, their use has come under severe criticism [6]. Research has shown that organic inhibitors are viable and highly beneficial because they are efficient, environmentally benign and comparatively cheap [7,8,9,10] and are more effective than inorganic compounds [11]. Benzothiazole-derived phthalonitriles and phthalocyanines (structures shown in Figure 1) are employed as organic inhibitors in this work. Corrosion inhibition using benzothiazole and its derivatives has been reported [12,13,14,15,16]. Compounds containing N and S in their structure such as benzothiazole have been reported to show excellent corrosion inhibition performance, since they have unshared electron pairs on N and S which are capable of forming bonds with metals. Hence benzothiazole and benzothiazole substituted phthalocyanines are employed in this work.

Phthalocyanines (Pcs) possess planar π-conjugated conformations [17], a property which enhances their ability to adsorb onto the metal surface and hence have been exploited for corrosion inhibition purposes [18,19,20,21]. This work reports on the corrosion inhibition of the phthalocyanine substituted with benzothiazole (BTThio) when compared to the latter alone. The Pc employed is tetrakis[(benzo[d]thiazol-2ylthio)phthalocyaninato]gallium(III)chloride (ClGaBTThioPc) (Figure 1). BTThio was involved in the synthesis of ClGaBTThioPc, therefore, it is expected that corrosion inhibition would be enhanced in the latter due to the π-conjugated electron rich Pc to which benzothiazole containing N and S in their structures are substituted. The presence of Ga(III) as a central metal allows for an attachment of an axial ligand, which is expected to reduce the aggregation of MPcs. Aggregation limits the application of MPcs in corrosion inhibition [19].

As stated above, small organic molecules such as benzothiazoles have been employed for corrosion inhibition. We show in this work that large π conjugated molecules such as phthalocyanines provide better protection against corrosion by comparing ClGaBTThioPc and its starting material, BTThio. This work provides proof that the extend of π conjugation affects corrosion inhibition. In addition, the uses of phthalonitrile derivatives as potential sources of developing corrosion inhibitors are under explored, hence a subject of this work when compared to the corresponding phthalocyanine. A detailed theoretical study was performed in this paper for BTThio and ClGaBTThioPc using density functional theory (DFT). The correlations between the inhibition efficiencies and the theoretical parameters such as the highest occupied molecular orbital (HOMO), lowest unoccupied molecular orbital (LUMO), energy gap (ΔE), global hardness (η), global softness (δ) and electronegativity (χ) has been discussed.

## 2. Results and Discussion

### 2.1. Open Circuit Potential (OCP) Time Evolution

Figure 2 presents the effect of the absence and presence of the studied inhibitors on the OCP of aluminum in 1.0 M hydrochloric acid solution. The OCP of aluminum was monitored in the presence of different inhibitor concentrations. A stabilization time of 30 min was allowed before the electrochemical measurements were performed and this time was deemed to be sufficient to attain a stable open circuit potential.

Shifts towards positive values of the OCP for metal dissolution processes in the presence of the inhibitors compared to OCP values in their absence are generally explained to be due to the influence on the anodic metal corrosion process [22]. In literature, the influence of the cathodic metal corrosion process is generally ascribed to shifts towards negative values in the presence of inhibitors [23,24].

Figure 2a,b, therefore, suggest that presence of BTThio and ClGaBTThioPc predominantly affect the cathodic aluminum corrosion in 1.0 M hydrochloric acid. This assertion was made due to the results presented in Figure 2a,b which show that OCP values of aluminum corrosion in the presence of the inhibitors (in most cases) are more negative compared to the OCP value in their absence suggesting predominantly cathodic character. ClGaBTThioPc gave more negative OCP values than BTThio showing more cathodic behavior for the former. ClGaBTThioPc possesses more electron rich properties due to its extended conjugation, hence, its predominant cathodic character. At the start in Figure 2, an abrupt shift of OCP towards positive potentials was noticed. This initial increase seems to be related to the formation and thickening of the oxide film on the metallic surface, suggesting corrosion protection [25].

### 2.2. Potentiodynamic Polarization Measurements

Polarization plots for BTThio and ClGaBTThioPc at different concentrations in 1.0 M HCl solution are presented in Figure 3.

Polarization measurements enabled the determinations of the following electrochemical parameters: corrosion current density (i_corr_), corrosion potential (E_corr_), and anodic (*β*_a_) and cathodic (*β*_c_) Tafel slopes and corrosion inhibition efficiency (IE%). The values of these parameters are shown in Table 1.

#### 2.2.1. Corrosion Current Density (i_corr_)

The corrosion current density (i_corr_) values decreased in the presence of studied corrosion inhibitors when compared to the blank solution (Table 1). Plausibly this is due to inhibition of Cl^−^ induced metal corrosion by the studied inhibitors and is consistent with findings for the two pyridinecarboxaldehyde thiosemicarbazone compounds studied as corrosion inhibitors [26]. Table 1 shows that the increase in inhibitor concentration generally caused a further decrease in corrosion current density. In all studied concentrations, in the presence of ClGaBTThioPc, corrosion current densities are smaller than those obtained in the presence of BTThio. This is an indication that the Pc is a better corrosion inhibitor.

#### 2.2.2. Corrosion Potential (E_corr_)

Yan and co-workers [26] classified a corrosion inhibitor as anodic or cathodic if the difference in corrosion potential (E_corr_) is more than 85 mV when compared to corrosion potential of the blank. In the presence of the corrosion inhibitors, the differences in the corrosion potential values are all less than 85 mV compared to the blank (Table 1) suggesting they are of a mixed-type character indicating possible retardation of anodic and cathodic aluminum corrosion processes. The lack of definite trend in the shift of E_corr_ values with change in concentration was ascribed to mixed-type character (anodic and cathodic) by Karthik and Sundaravadivelu [27] for atenolol as corrosion inhibitor. The same applies in this work.

#### 2.2.3. Tafel Slopes (*β*_a_ and *β*_c_)

Values of cathodic Tafel slopes (*β*_c_), presented in Table 1, are all higher (more positive) in the presence of corrosion inhibitors than in their absence suggesting a predominant cathodic character. Predominant cathodic character for the studied corrosion inhibitors can be inferred based on the consideration that *β*_c_ values are larger (in magnitude) than *β*_a_ values, except for in the presence of 8 µM ClGaBTThioPc. Similar observations and conclusions were reported earlier for 1-hexyl-3-methylimidazolium-based ionic liquids and some phenolic compounds investigated as corrosion inhibitors [28,29].

#### 2.2.4. Corrosion Inhibition Efficiency (IE%)

The corrosion inhibition efficiency values (IE%) were calculated using Appendix A, and are presented in Table 1. Values of IE% presented in Table 1 increased with the concentration of the inhibitors (with the exception of 10 µM ClGaBTThioPc), these indicate that the presence of the inhibitors retarded the metal corrosion process. It is generally believed that metal surfaces are protected by corrosion inhibitors via adsorption of the inhibitor molecules onto metal surfaces [23,26]. The corrosion inhibition efficiencies are higher for ClGaBTThioPc, showing superior inhibition.

### 2.3. Adsorption Isotherms

El-Awady, Freundlich, Frumkin, Langmuir and Temkin isotherm linear plots [5,30,31] are presented in Figure 4 for adsorption of the studied inhibitors on aluminum surface in 1.0 M hydrochloric acid solution (see Appendix A). Considering that adsorption of inhibitor molecules onto the metal surface is the first step in the mechanism of corrosion inhibition [32], it is important to understand the adsorption behavior of organic molecules (inhibitors) on the metal surface. Thus, the adsorption of the studied corrosion inhibitors was investigated by subjecting the experimentally calculated surface coverage (θ) values to various adsorption isotherms such as El-Awady, Langmuir, Temkin, Frumkin and Freundlich isotherms. The values of equilibrium constants from the isotherm plots: Langmuir (K_L_), Freundlich (K_F_), Frumkin (K_Fr_), Temkin (K_T_), and El-Awady (K_El_) are presented in Table 2.

Langmuir adsorption isotherm was derived based on the assumption that the adsorption surface possesses sites with equivalent/similar energies of adsorption, allowing maximum/saturated adsorption of solute molecules for adsorbent monolayer coverage [33]. It implies that molecules following Langmuir adsorption mechanism should have slope value of unity as predicted by Appendix A. The linear form of Langmuir adsorption isotherm used herein (see Appendix A) has C as the inhibitor concentration, θ is surface coverage (θ = IE%/100) [7] and K_L_ is the adsorption equilibrium constant obtained from Langmuir isotherm. Values of regression coefficient (R^2^) 0.9871 for BTThio and 0.9971 for ClGaBTThioPc were obtained from Langmuir isotherm plots, Figure 4a, hence the Pc shows better fit. The values of K_L_ were derived from the intercept of the plots, and the K_L_ values for BTThio and ClGaBTThioPc are 5.0 × 10^5^ M^−1^ and 1.0 × 10^6^ M^−1^ respectively; these are large and positive indicating strong adsorption onto the metal surface. The Langmuir isotherm (Appendix A) predicts a slope value of unity but 1.2 and 1.1 M are obtained. Though these values are very close to unity, values of Chi-square statistic presented in Table 2 do not give the smallest error values. By implication, the Langmuir isotherm does not give the best explanation to the adsorption profiles of the studied inhibitors onto the metal surface. The theoretical derivation of the Langmuir isotherm assumes that the adsorbent (herein the metal) surface is homogeneous, implying that the adsorption surface sites should possess similar energy as discussed above.

SEM images (Figure 5a), reveal that the aluminum surface is non-homogeneous, this gives an insight that the metal adsorption sites are heterogeneous because they most probably have different energies of adsorption. This explains the inability of the Langmuir isotherm to fit inhibitor’s adsorption profiles and further supported by Figure 6 which compares experimental and theoretically determined inhibition efficiency values of the inhibitors.

The Freundlich isotherm models adsorption of molecules on heterogeneous surfaces and the linear form used herein (see Appendix A) has C as the inhibitor concentration, θ is surface coverage (calculated as described above), “n” is Freundlich heterogeneity constant and K_F_ is the adsorption equilibrium constant obtained from Freundlich isotherm. Respectively, the values of K_F_ and “n” for the studied compounds were derived from the intercepts and slopes of Figure 4b plots (Appendix A) for the studied molecules. Freundlich isotherm plots gave R^2^ values of 0.9728 and 0.9725 for adsorption data plots of BTThio and ClGaBTThioPc respectively, showing similar fits. The equilibrium constant (K_F_) calculated from the Freundlich plots are 13.1 and 7.7 M^−1^, respectively for BTThio and ClGaBTThioPc. Thermodynamically, equilibrium constant value less than unity suggests unfavorable process; equal to zero suggests the process is in dynamic equilibrium and more than unity suggests a favorable process. Thus the equilibrium constant values obtained for the studied inhibitors suggest they adsorbed favorably onto the metal surface. The slope of the Freundlich plot (Figure 4b, Appendix A) allows the determination of an adsorption parameter (n) which gives a measure of the inhibitor adsorption layer [8]. Values of the Freundlich adsorption parameter (n) were calculated as 3.9 and 5.1 M respectively for the adsorption of BTThio and ClGaBTThioPc on the metal surface, suggesting multilayer adsorption which is consistent with physisorption. The incorporation of benzothiazole moiety (BTThio) in a Pc ring to give ClGaBTThioPc, therefore, did not cause a change in adsorption mechanism. Chi-square statistics gave smaller χ^2^ values for adsorption data of ClGaBTThioPc compared to BTThio when fitted to Freundlich.

The Temkin isotherm model is based on adsorption of molecules on heterogeneous surfaces with adsorption sites possessing logarithmic energies from the equation reported earlier [34]. The linear form used herein (see Appendix A) has C as the inhibitor concentration, θ is surface coverage (calculated as described above), “f” is Temkin heterogeneity constant and K_T_ is the adsorption equilibrium constant obtained from Temkin isotherm. The values of K_T_ and “f” were derived from the intercepts and slopes of Figure 4c plots (Appendix A) respectively for the studied molecules. Temkin equilibrium constant (K_T_) values calculated for BTThio and ClGaBTThioPc are 1.07 × 10^7^ and 4.1 × 10^7^ M^−1^, respectively, these values are large and suggest favorable adsorption of the inhibitors on the metal surface in 1.0 M hydrochloric acid solution. Temkin adsorption parameters (f) of 6.8 and 7.4 M, respectively, for BTThio and ClGaBTThioPc are more than unity and positive suggesting the presence of attractive molecular interactions [35], supporting multilayer adsorption of the inhibitor molecules due to physisorption, as was the case with Freundlich adsorption discussed above. Chi-square statistics gave smaller χ^2^ values for adsorption data of ClGaBTThioPc compared to BTThio when fitted to Temkin isotherm.

The adsorption model of the El-Awady isotherm characterizes the adsorption sites on the surface [36] and the linear form used (see Appendix A) has C as the inhibitor concentration, θ is surface coverage, “Y_El_” is El-Awady heterogeneity constant and K_El_ is the adsorption equilibrium constant obtained from El-Awady isotherm. The values of K_El_ and “Y_El_” were derived respectively from the intercepts and slopes of Appendix A plots for the studied molecules. El-Awady plots for the adsorption data of the studied inhibitors onto the metal in 1.0 M hydrochloric acid solution are presented in Figure 4d. The equilibrium constants from the El-Awady plots (K_El_) were calculated from the intercepts of the plots to give 2.5 × 10^3^ and 1.0 × 10^4^ M^−1^, respectively for BTThio and ClGaBTThioPc, respectively. These values are more than unity suggesting favorable adsorption. BTThio and ClGaBTThioPc have El-Awady adsorption parameter (Y_El_) values calculated from slopes of El-Awady plots as 0.6 M^−1^ and 0.7 M^−1^, respectively. These intrinsic adsorption parameters give a measure of the adsorbed inhibitor layers [8], suggesting that they are less than one inhibitor molecule thick on the metal surface, which is characteristic of chemisorption. As was the case with adsorption data fitted to Freundlich and Temkin isotherms, the χ^2^ value is smaller for ClGaBTThioPc compared to BTThio for El-Awady isotherm.

The Frumkin adsorption model is an isotherm type which explains the extent of lateral interactions among the adsorbed molecules on adsorbent surface [33], the linear form used herein (see Appendix A), where C is the inhibitor concentration, θ is surface coverage, “*α*” is the Frumkin lateral interaction constant describing the interaction in adsorbed layer and K_Fr_ is the adsorption equilibrium constant obtained from Frumkin isotherm. The values of K_Fr_ and “*α*” were derived respectively from the intercepts and slopes of Figure 4e,f (Appendix A) for the studied molecules. The equilibrium constants from the Frumkin plots (K_Fr_) were calculated from the intercepts of the plots to give 3.4 × 10^−7^ M and 1.5 × 10^−7^ M for BTThio and ClGaBTThioPc respectively. These values are less than unity which depict unfavorable adsorption. BTThio and ClGaBTThioPc however have Frumkin adsorption parameter (*α*) values calculated from slopes of Frumkin plots as 1.6 M and 2.5 M, respectively. These intrinsic adsorption parameters are positive suggesting attractive lateral interactions of the adsorbed inhibitor molecules [31]. As was the case with adsorption data fitted to Freundlich and Temkin isotherms, the χ^2^ value is smaller for ClGaBTThioPc compared to BTThio for Frumkin isotherm.

Inhibition efficiency values versus concentration plots for El-Awady, Freundlich, Frumkin, Langmuir and Temkin isotherms when compared with experimentally determined values are presented in Figure 6. El-Awady, Freundlich and Temkin isotherms gave very good descriptions of BTThio and ClGaBTThioPc adsorption on aluminum in 1.0 M hydrochloric acid solution as shown in Figure 6, while Frumkin and Langmuir were not appropriate.

Free energies of adsorption ($\Delta G_{ads}^{0}$) values were calculated from the various equilibrium constants (see Appendix A), and these values are presented in Table 2. Values of $\Delta G_{ads}^{0}$ around −20 kJmol^−1^ or less (in magnitude) are consistent with physisorption, and those around −40 kJmol^−1^ or more (in magnitude) are due to chemisorption as reported earlier [1]. BTThio has $\Delta G_{ads}^{0}$ values as follows: −42.9 kJ/mol (Langmuir), −16.5 kJ/mol (Freundlich), +27.2 kJ/mol (Frumkin), −50.6 kJ/mol (Temkin) and −29.6 kJ/mol (El-Awady). In a similar manner $\Delta G_{ads}^{0}$ values for ClGaBTThioPc are as follows: −44.6 kJ/mol (Langmuir), −15.2 kJ/mol (Freundlich), +29.3 kJ/mol (Frumkin), −53.9 (Temkin) and −33.1 kJ/mol (El-Awady).

Langmuir and Temkin isotherms predict chemisorption for the studied inhibitors on aluminum in 1.0 M hydrochloric acid solution because the calculated $\Delta G_{ads}^{0}$ values are more than −40 kJ/mol. Physisorption is depicted to be followed by Freundlich isotherm for the studied inhibitors considering that $\Delta G_{ads}^{0}$ values less than −20 kJ/mol were calculated. $\Delta G_{ads}^{0}$ values of the studied inhibitors are between −20 kJ/mol and −40 kJ/mol from calculations made using El-Awady parameters, suggesting competitive physisorption and chemisorption. Negative signs of the calculated values support spontaneous processes therefore favorable adsorption of the inhibitors. Thus Frumkin isotherm shows unfavorable adsorption with a positive value. Different adsorption mechanisms predicted from $\Delta G_{ads}^{0}$ values for the studied inhibitors, suggest that corrosion inhibition is a complex process. This assertion is supported by findings reported for the following corrosion inhibitors: cashew nut testa tannin, pyridinium-ionic liquid, isatin derivatives, clozapine and pyridine Schiff base derivatives [37,38,39,40,41].

### 2.4. Electrochemical Impedance Spectroscopy (EIS)

Nyquist plots are presented in Figure 7 for aluminum in 1.0 M hydrochloric acid solution with and without BTThio and ClGaBTThioPc. Large capacitive curves are apparent at high frequency followed by inductive curves at low frequency values. In the presence of the inhibitors, the capacitive curve diameters are larger than that of the blank solution and increases with inhibitor concentration, with those of ClGaBTThioPc being larger than those of BTThio. This indicates that the presence of the inhibitors cause an increase in the impedance of the inhibited substrate in a manner that those containing ClGaBTThioPc are increased more. Similar analogy was posited earlier by Li et al. [42].

The capacitive curves are often attributed to the charge transfer of the corrosion process. Figure 7a,b show the Nyquist plots that have depressed semicircle characteristic of capacitive curves explained to be caused by the dispersion effect from surface irregularities and heterogeneities [43]. The inductive curves at low frequency values are thought to be caused by the relaxation process when species such as $H_{ads}^{+}$ or inhibitor species adsorb on the electrode surface [42]. The inductive curve in Nyquist plots (Figure 7) therefore may be closely related to the existence of a passive film on aluminum [44]. Inductive curves seen in the presence of the inhibitors are larger than in their absence, suggesting a significant role of inhibitor species adsorption onto aluminum.

Analyses of the impedance data were performed using equivalent circuit shown in Figure 7c. The solution resistance (R_s_), charge transfer resistance (R_t_), constant phase element (CPE), are shown in Figure 7c; the inductive resistance (R_L_) and inductance (L) are introduced to simulate the inductive curves. The inhibitor efficiency (IE%) values were calculated using the Appendix A. Selected impedance parameters and inhibition efficiency values from the fitting of the EIS data are presented in Table 3. The CPE has an exponent, “n”, used to study the changes on the metal/solution interface. Values of n close to unity are due to frequency dispersion caused by arbitrary current distribution on electrode surface [42,43] indicating the predominance of capacitive behavior [45], as is the case in this work. Large R_t_ values are associated with slower corroding process [40], consequently, in the presence of studied inhibitors aluminum corrosion slows down more at increased concentration. The values of inhibition efficiency calculated from EIS are in good agreement with those determined from polarization curves.

### 2.5. Surface Analysis

#### 2.5.1. FTIR Spectra

This FTIR spectra of studied inhibitors (BTThio and ClGaBTThioPc), corroded aluminum in the absence and presence of the inhibitors are shown in Figure 8. Vibrational peak around 3100 cm^−1^ seen in the infrared spectrum for corroded aluminum (labelled as Al_corr_ in Figure 8) can be ascribed to the signals of aluminum hydroxide Al(OH)_3_, AlOOH and hydrated aluminum (Al-H_2_O) [46]. This indicates that metal surface composed of oxides of aluminum that form a protective thin film. Interestingly the infrared spectrum of corroded aluminum in the presence of BTThio (labelled as BTThio_corr_) shows a reduction in the intensity of the peak at 3100 cm^−1^ (shifted to 3300 cm^−1^), Figure 8. In a similar manner, the infrared spectrum of corroded aluminum in the presence of ClGaBTThioPc shows a further decrease in the intensity of the peak at 3100 cm^−1^ (shifted to 3300 cm^−1^), and this provides evidence of improved metal corrosion retardation than BTThio.

Vibrational signals characteristic of phthalonitriles and ascribed to C-N moieties [47] appeared in the spectrum of BTThio at 2231 cm^−1^ (before corrosion) as Figure 8 shows and this peak disappeared after corrosion, suggesting the involvement of the C–N groups as adsorption sites for BTThio onto the metal surface. Vibrational signals at 1534 cm^−1^ (–CN), 1424 cm^−1^ (–CH_2_), 1279 cm^−1^ (C–H), 1077 cm^−1^ (C=C and –CN) and 910–720 cm^−1^ (aromatic C=C and C–H) have been ascribed earlier [7,48,49,50]. After corrosion, these vibrational signals disappeared which indicates good BTThio/aluminum adsorption interaction. The assignment of 751 cm^−1^ to C–S–C vibration is consistent with the report of Omaka and co-workers [51] for benzothiazole moiety. The disappearance of this signal after corrosion is a good indication that corrosion inhibition by BTThio is due to adsorption onto metal surface. On comparison, the spectra of BTThio (before corrosion) and BTThio_corr_ (after corrosion) possess marked differences in the afore-mentioned infrared vibrational regions, suggesting that the BTThio molecule was effectively adsorbed onto the metal and led to corrosion inhibition via inhibitor/metal adsorption. Formation of BTThio-aluminum complex as thin film covering metal surface is thus evidenced by FTIR.

The infrared spectra of ClGaBTThioPc (before) and ClGaBTThioPc_corr_ (after) in the absence and presence of aluminum in 1.0 M hydrochloric acid solution respectively, are shown in Figure 8. Vibrational signals seen in the 1796–2647 cm^−1^ range ascribed to Ar-CN vibrations [52,53,54], appear after metal corrosion in the presence of ClGaBTThioPc as Figure 8 presents. Infrared spectra of corroded aluminum and ClGaBTThioPc (before corrosion) show that these vibrational signals were absent, revealing the role of the tetra-condensed porphyrazine structure of ClGaBTThioPc in the corrosion inhibition process. These indicate that ClGaBTThioPc possesses very high electron density due to the presence of de-localized electrons in the rich (condensed) aromatic structure shown in Figure 1, which imparts on it good corrosion inhibition character. Formation of ClGaBTThioPc-aluminum complex as thin film covering metal surface is thus evidenced by FTIR.

Herein, an attempt has been made to estimate corrosion inhibition efficiency using FTIR peak intensities (transmittance values). Using the technique of FTIR peak intensities (transmittance values) and bearing in mind the relationship between transmittance and absorbance (Equation (1)), then, Equation (2) applies:Absorbance (A) = 2 − log(%T)(1)
(2)$$\mathrm{IE}\%=\frac{A_{0}-A_{x}}{A_{0}}\times 100$$
where *A*_0_ represents the absorbance of the infrared peak (at 3300 cm^−1^) for corroded aluminum in the absence of inhibitor, *A_x_* represents the absorbance of the infrared peak for corroded aluminum in the presence of an inhibitor. The inhibition performance values of 98.2% and 99.5% respectively for BTThio and ClGaBTThioPc indicating that ClGaBTThioPc protected metal surface better than BTThio in hydrochloric acid solution.

#### 2.5.2. SEM and EDX

Figure 5 shows the SEM image of uncorroded aluminum, corroded aluminum in the absence of inhibitors and corroded aluminum in the presence of inhibitors. Earlier work [55] used SEM to characterize the heterogeneous nature of aluminum surface, revealing the microstructure propeties of the metal surface. Rosa and co-workers [56] reported the correlation between surface structure and corrosion resistance and their findings are supported by the work of Osorio and co-workers [57]. To this end therefore, the heterogeneity of the uncorroded metal surface adsorption sites is revealed in the SEM image presented in Figure 5a. Figure 5b reveals that after immersion in hydrochloric acid in the absence of inhibitor, the metal surface becomes corroded badly due to the attack of chloride ion. The formation of a white colored “flower-like” substance thought to be from the corrosion resistant aluminum oxide film, failed to protect the metal from the attack of chloride ion which agrees with an earlier report [58]. The corrosion attack on the metal surface decreased markedly in the presence of BTThio, Figure 5c, a strong indication that BTThio presence protected the metal surface from Cl^−^ attacks effectively, hence, its use for aluminum corrosion inhibition is encouraged. In the presence of ClGaBTThioPc, shown in Figure 5d, the metal surface was effectively protected in the presence of the inhibitor, showing marked improvement over BTThio protection performance at the same inhibitor concentration.

Very high aluminum composition was revealed by energy dispersive X-ray (EDX) measurement for the uncorroded metal (Appendix A) and small contents of carbon and oxygen. In the absence of the inhibitors, EDX shows very high contents of chlorine and oxygen, Appendix A. Appendix A reveals the absence of chloride ion on the metal surface in the presence of ClGaBTThioPc, suggesting effective protection of the metal from aggressive attacks of Cl^−^. The absence of chloride ion in the presence of ClGaBTThioPc has been reported earlier for the use of phthalocyanines as corrosion inhibitors [20]. There is still some small amount of chloride ions for BTThio (Appendix A). The EDX results therefore strongly support that the presence of Cl^−^ cause enhanced metal corrosion and also the presence of electron rich compounds play effective roles by adsorbing on the metal surface, retarding metal corrosion. The metal surface protection was feasible possibly due to the formation of a film on the metal surface, leading to effective metal protection in hydrochloric acid solution by mainly ClGaBTThioPc. The oxygen contents of the corroded metal surfaces clearly decreased in the presence of ClGaBTThioPc, reflecting better corrosion inhibition than BTThio.

#### 2.5.3. X-ray Diffraction Studies

The XRD patterns of the uncorroded aluminum surface, corroded aluminum surfaces in 1.0 M hydrochloric acid in the absence and presence of BTThio or ClGaBTThioPc are presented in Figure 9. Uncorroded aluminum shows peak patterns at about 2θ = 39.3, 45.5, 65.8, 78.8 and 83.1°. These peaks correspond to aluminum signals of gibbsite [*γ*-Al(OH)_3_], bayerite [*α*-Al(OH)_3_], alumina (Al_2_O_3_), boehmite (AlOOH) and metastable alumina (*χ*-Al(OH)_3_ and *κ*-Al(OH)_3_) [59,60,61,62]. These diffraction peaks are indexed as orthorhombic phase of boehmite AlOOH (JCPDS number 00-021-1307) for 2θ = 39.3 and 65.8°, as cubic γ-Al_2_O_3_ (JCPDS number 00-001-1303) for 2θ = 45.5° and as face centered cubic aluminum (JCPDS number 004-0787) for 2θ = 65.8 and 78.8°. On comparison with diffraction peaks of metallic aluminum, peaks at 2θ = 39.3, 45.5, 65.8, 78.8 and 83.1° match well (JCPDS number 89-4037) [63,64,65,66,67].

Figure 9 presents the XRD patterns of corroded Al in the absence of inhibitors (Al_corr_) with peaks at 2θ = 39.8, 41.5, 46.0, 65.7, 78.8 and 99.6°, revealing two new peaks at 2θ = 41.5 and 99.6° for corroded Al and the disappearance of the peak for uncorroded Al at 2θ = 83.1°. Following protection with inhibitors (ClGaBTThioPc and BTThio), the latter peak reappears.

Herein, therefore, a successful attempt has been made to estimate corrosion inhibition efficiency using XRD peak heights by the following Equations (3) and (4):(3)$$IR=\frac{H_{wi}}{H_{w}}$$
(4)$$\mathrm{IE}\%=\frac{IR_{wi}-IR_{w}}{IR_{wi}}\times 100$$
where *IR* represents relative corrosion rate, *H_wi_* and *H_w_* respectively stand for peak height (2θ = 99.7°) of corroded metal in the presence of an inhibitor and in the absence of an inhibitor, *IR_wi_* and *IR_w_* are for the relative corrosion rates of the metal with inhibitor and without respectively. Calculated corrosion inhibition efficiency values are 51.3% and 70.3%, respectively for corroded metal in the presence of BTThio and ClGaBTThioPc which corroborates trend of inhibition efficiency values calculated from the polarographic technique.

### 2.6. Quantum Chemical Studies

The optimised structures of the studied corrosion inhibitors are shown together with the corresponding electron density surfaces of the highest occupied molecular orbitals (HOMO) and lowest unoccupied molecular orbitals (LUMO) of the molecules in Figure 10.

The HOMO is localized mainly on the benzothiazole moiety, and the para-substituted thio-part of the phthalonitrile, suggesting that the adsorptive interactions of BTThio with the metal surface via π-electron donation to the empty d-orbitals predominantly occur in this region. The HOMO orbitals around the S-atom of the para-substituted thio-part of the phthalonitrile are σ-type, indicating that this group can interact with the empty p-orbitals of the metal. In contrast, the LUMO is localized mainly on the phthalonitrile, and the para-substituted thio-part of the phthalonitrile suggesting that this region of BTThio molecule can accept a charge from occupied orbitals of a metal. The HOMO and LUMO of ClGaBTThioPc are localized mainly on the phthalocyanine ligand and the benzylic part of the benzothiazole substituents.

The following molecular parameters: the HOMO energy (E_HOMO_), the LUMO energy (E_LUMO_), the energy gap (ΔE), global hardness (η), global softness (δ), global electronegativity (χ), and fraction of electron transferred from inhibitor to metal (ΔN); are derived from the calculations for the studied inhibitor molecules and presented in Table 3. The derived parameters, ΔE (ΔE = E_LUMO_ − E_HOMO_), η, χ, and ΔN were calculated as previously reported [46] and were helpful in explaining the performance of compounds as corrosion inhibitors [8].

Interactions which occur when corrosion inhibitors adsorb on aluminum surface is such that the electron donating abilities of the inhibitor molecules to the metal surface (acceptor) is enhanced by high E_HOMO_ value. Calculated E_HOMO_ values for the inhibitors, as shown in Table 3, reveal that ClGaBTThioPc has the higher E_HOMO_ which is consistent with its better inhibition efficiency and corresponds to enhanced donor ability. ClGaBTThioPc has lower E_LUMO_ value than BTThio supporting better electron accepting ability from the metal, which improves adsorption property of inhibitor molecule onto the metal and translates to higher inhibition efficiency. The adsorption and corrosion inhibition properties of an organic compound are related to donor-acceptor abilities which favor both forward and backward donation of electrons and appear to affect inhibitive potential as aforementioned for the calculated E_HOMO_ and E_LUMO_ values.

Energy gap (ΔE), global hardness (η), global softness (δ) and electronegativity (χ) values calculated can be used to assess the relative reactivity of molecules. Results presented in Table 3 indicate that ClGaBTThioPc has higher reactivity than BTThio, hence better corrosion inhibition performance, because of its lower values of ΔE and η and higher value of δ. The lower value of χ for ClGaBTThioPc translates to a poor grip of molecular electrons and corresponds to the better tendency of releasing electrons during intermolecular interactions. Nnaji and co-workers [5] earlier reported that higher ΔN (less negative) value suggest better electron donating ability, a property known to retard metal oxidation process. From the foregoing, higher ΔN value which corresponds to better corrosion inhibition performance of metallated phthalocyanine inhibitor corroborates higher inhibition performance presented in Table 4. Negative ΔN value was calculated for an aluminum corrosion inhibitor, ethylenediaminetetraacetic acid (EDTA), and the report posited that EDTA donated electrons to the metal surface [68]. ΔN value less than 3.6 has been reported to affect the electron donation by the inhibitor in a way that increased electron donation to the metal surface increase inhibition efficiency [69,70,71].

### 2.7. Inhibition Mechanism

Equations describing the general mechanism for the dissolution of aluminum have been reported [72]. Adsorption of inhibitor molecules consequently leads to retardation of metal corrosion by blocking the formation of AlOH_ads_ species available for the rate determining steps of the oxidation reaction and the complexation of chloride ions. As earlier reported by Obi-Egbedi and Obot [73], in the presence of inhibitors (TThio or ClGaBTThioPc), adsorption onto the metal surface is generally believed to occur, as proposed in Figure 11, protecting the metal from the aggressive attack of Cl^−^.

In agreement with earlier works [2,74,75], the electron-rich centers of the inhibitor such as lone electron pairs of heteroatoms and delocalized electrons of unsaturated and aromatic parts of the inhibitor molecules; are the adsorption sites favoring inhibitor molecule adsorption onto the metal surface.

## 3. Experimental Section

### 3.1. Materials

Aluminum coupons (>99%) were cut into 2 × 2 × 0.1 cm sizes from metal sheets. The area of the working electrodes exposed to the electrolyte was 3.0 cm^2^. Hydrochloric acid (Merck, Kenilworth, NJ, USA) used was of the highest available purity (>99%). Ultra-pure water was obtained from a Milli-Q Water System (Millipore Corp., Burlington, MA, USA). Solvents used were of the highest available purity (>99%) and were supplied by Sigma Aldrich (St. Louis, MI, USA), Fluka (St. Louis, MI, USA) and Merck. 4-(Benzo[d]thiazol-2ylthio)- phthalonitrile (BTThio) and tetrakis[(benzo[d]thiazol-2ylthio)phthalocyaninato]gallium(III)chloride (ClGaBTThioPc) were synthesized as reported before [76] and used as corrosion inhibitors.

### 3.2. Equipment

Scanning electron microscopy (SEM) images were obtained using a JSM 840 scanning electron microscope (JEOL, Tokyo, Japan). Elemental compositions were qualitatively determined using an INCA PENTA FET energy dispersive X-ray spectroscopy (EDX) instrument (Oxford Instruments, High Wycombe, UK), operated at 20 kV accelerating voltage. FTIR spectra of the films were obtained on an Alpha IR (100 FT-IR) spectrophotometer (Bruker, Billerica, MA, USA) with a universal attenuated total reflectance (ATR) sampling accessory. X-ray diffraction (XRD) patterns were recorded using Bruker D8 Discover equipped with a Lynx Eye detector, using Cu-Ka radiation (λ = 1.5405 Å, Nickel filter).

For XRD and FTIR analyses, the adsorbed films were carefully removed by scraping with a clean stainless steel blade, and care was taken to avoid contamination during the collection of the corrosion products was avoided. For SEM/EDX analyses the electrode surfaces were captured/accessed directly.

All the electrochemical experiments were carried out using a BAS 100B electrochemistry setup (West Lafayette, IN 47906 USA) except for electrochemical impedance spectroscopy (EIS) studies which were performed using an PGSTAT 30 potentiostat (Autolab, KM Utrecht, The Netherlands) equipped with GPES software version 4.9.

### 3.3. Electrochemical Measurements

The inhibitors (BTThio and ClGaBTThioPc) are not water soluble, hence their solutions were first prepared by weighing the required amount and dissolving in 10 mL tetrahydrofuran (THF) then making up to 100 mL using 1.0 M hydrochloric acid to give a 10 µM solution. The rest of the concentrations were prepared by serial dilution from the 10 µM solution. The concentrations employed are 2, 4, 6, 8, and 10 µM. Aluminum coupons of 3.0 cm^2^ exposed area were used as working electrodes and were exposed to the electrolyte (1.0 M hydrochloric acid) in the absence and presence of the inhibitors). The temperature of the experiments was regulated at 28 °C by a thermostated laboratory hot plate made by Corning (model 6796-220, Corning, NY, USA) with a thermometer fitted to the reaction vessel.

A three-electrode electrochemical cell containing aluminum coupons, a platinum wire, and an Ag/AgCl electrode (in 3.0 M KCl) as the working, counter and reference electrodes, respectively, was employed. Before the metal coupons were immersed in the chloride solution, the surfaces of the working electrodes were abraded with emery papers of 400 and 800 grit sizes, washed in acetone, Millipore water and then dried.

Potentiodynamic polarization curves were obtained by scanning the potential from −1.0 V to 0.0 V vs. Ag/AgCl. These potential values was employed since the potentiodynamic polarization curves for Al occur in this range [3]. The open circuit potential (OCP) experiments were carried out after stabilization time of 30 min and the open circuit potentials (E_OCP_) were determined for the metal/HCl systems with and without the corrosion inhibitors. Corrosion parameters such as corrosion potential (E_corr_), corrosion current density (i_corr_), cathodic (*β*_c_) and anodic (*β*_a_) Tafel slopes were calculated. The inhibition efficiency (IE%) was calculated using the equation reported before [77] (see Appendix A). EIS measurements were performed at the OCP between 0.1 Hz and 10 kHz, using a 5 mV root-mean-square (rms) sinusoidal modulation. A non-linear least squares (NLLS) method based on the EQUIVCRT program was used for automatic fitting of the obtained EIS data. The measurements were performed in duplicate and the results are presented in the relevant tables. The errors reported in this work are comparable and even lower in some cases than those reported in literature [78,79,80].

### 3.4. Quantum Chemical Studies

The optimized structures of the studied corrosion inhibitors were obtained using the Density functional theory (DFT) technique. B3LYP is a hybrid functional comprising the Becke’s three parameter exchange functional [81] and Lee-Yang-Parr correlation functional [82,83], used in conjunction with SDD basis sets contained in the Gaussian 09 software package [84]. The calculated parameters include the energy of the highest occupied molecular orbital (E_HOMO_), the energy of the lowest unoccupied molecular orbital (E_LUMO_), the HOMO−LUMO energy gap (ΔE), the global softness (σ), the global hardness (η), the fraction of electrons transferred (ΔN) and the electronegativity (χ).

## 4. Conclusions

Cyclic voltammetry, open circuit potential and potentiodynamic polarization were used to study the aluminum corrosion inhibition potentials of benzothiazole derived phthalonitrile (BTThio) and phthalocyanine (ClGaBTThioPc) in 1.0 M hydrochloric acid solution. Open circuit potential values of aluminum corrosion in the presence of the suggest predominantly cathodic character with higher corrosion efficiency for ClGaBTThioPc compared to BTThio due to the higher π conjugation in the former. Adsorption studies revealed that BTThio and ClGaBTThioPc adsorption onto aluminum surface is complex in nature and followed Freundlich, Temkin and El-Awady isotherms. Electrochemical impedance spectroscopy confirmed that BTThio and ClGaBTThioPc inhibit the corrosion rate of aluminum by an adsorption mechanism. Effective metal surface protection was revealed with scanning electron microscopy (SEM) measurements in the presence of the inhibitors and energy dispersive X-ray measurements reveal that the most probable way by which the inhibitors protected the metal surface was by shielding it from the corrosion attacks of Cl^−^. Theoretical considerations suggest that the corrosion inhibition exhibited by BTThio and ClGaBTThioPc is by adsorption onto aluminum. Quantum chemical calculations strongly support the experimental results that demonstrate that the corrosion inhibition performance of the inhibitors is in the order ClGaBTThioPc > BTThio thus supporting the conclusion that inhibitors containing more unsaturation in their molecular structures perform better.

## Figures and Tables

**Figure 1 molecules-24-00207-f001:**
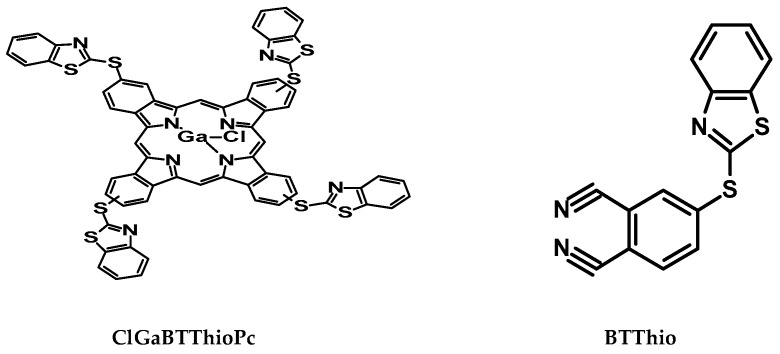
Structures of 4-(benzo[d]thiazol-2-ylthio)phthalonitrile (BTThio) and tetrakis[(benzo[d]- thiazol-2-ylthio)phthalocyaninato] gallium(III) chloride (ClGaBTThioPc).

**Figure 2 molecules-24-00207-f002:**
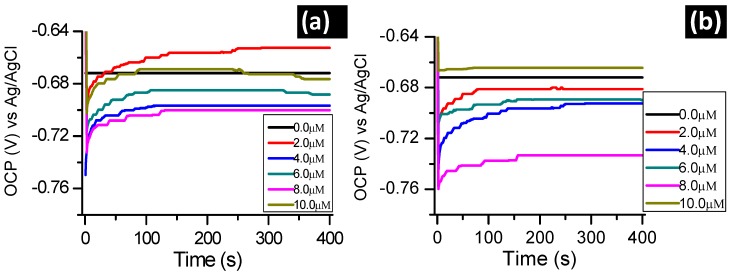
OCP evolution with time for aluminum in 1.0 M hydrochloric acid in the absence and presence of (**a**) BTThio and (**b)** ClGaBTThioPc.

**Figure 3 molecules-24-00207-f003:**
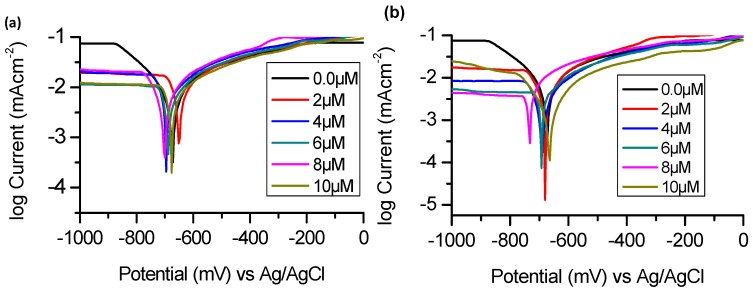
Potentiodynamic polarization curves for aluminum in 1.0 M hydrochloric acid in the absence and presence of (**a**) BTThio and (**b**) ClGaBTThioPc at 28 °C.

**Figure 4 molecules-24-00207-f004:**
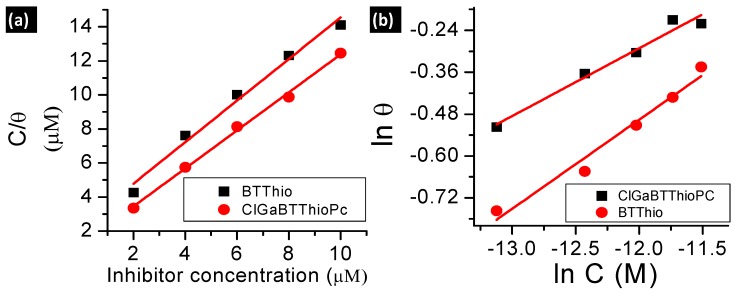
(**a**) Langmuir; (**b**) Freundlich; (**c**) Temkin; (**d**) El-Awady; (**e**,**f**) Frumkin adsorption (for BTThio and ClGaBTThioPc, respectively) isotherm plots at 28 °C.

**Figure 5 molecules-24-00207-f005:**
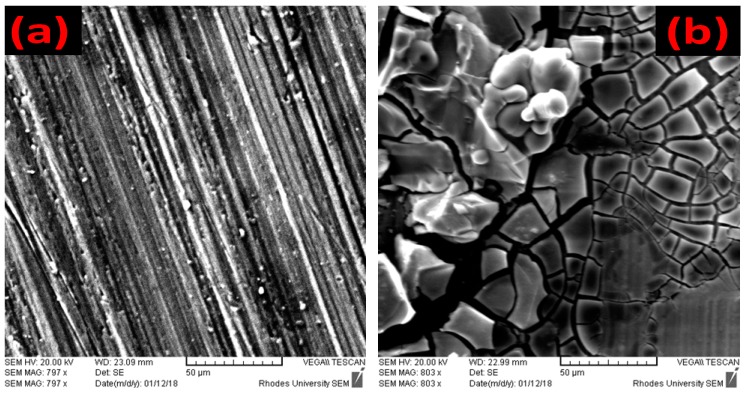
SEM images of (**a**) bare aluminum metal; (**b**) aluminum metal after immersion in 1.0 M HCl; (**c**) aluminum metal after immersion in 1.0 M HCl containing BTThio; and (**d**) aluminum metal after immersion in 1.0 M HCl containing ClGaBTThioPc.

**Figure 6 molecules-24-00207-f006:**
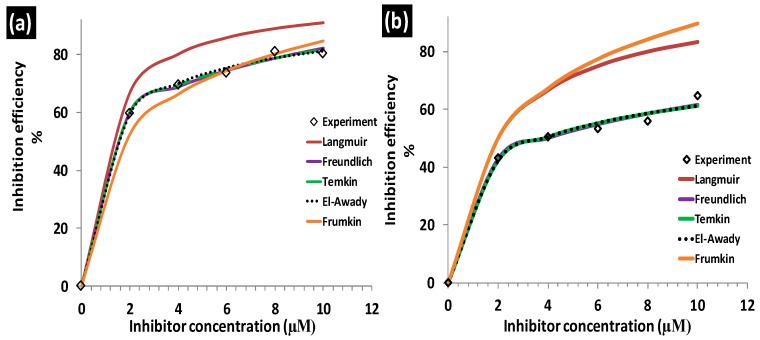
Comparison of experimental and theoretical inhibition efficiency values of (**a**) ClGaBTThioPc and (**b**) BTThio.

**Figure 7 molecules-24-00207-f007:**
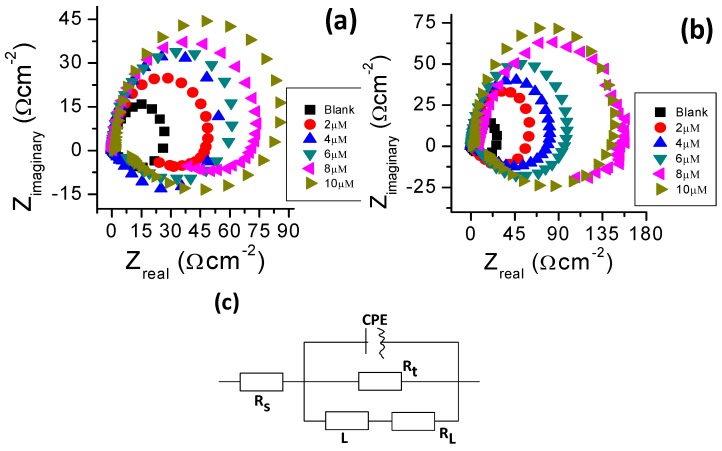
Nyquist plots of aluminium in 1.0 M hydrochloric acid solution with BTThio (**a**) with ClGaBTThio (**b**) and equivalent circuit used to fit EIS data for (**c**).

**Figure 8 molecules-24-00207-f008:**
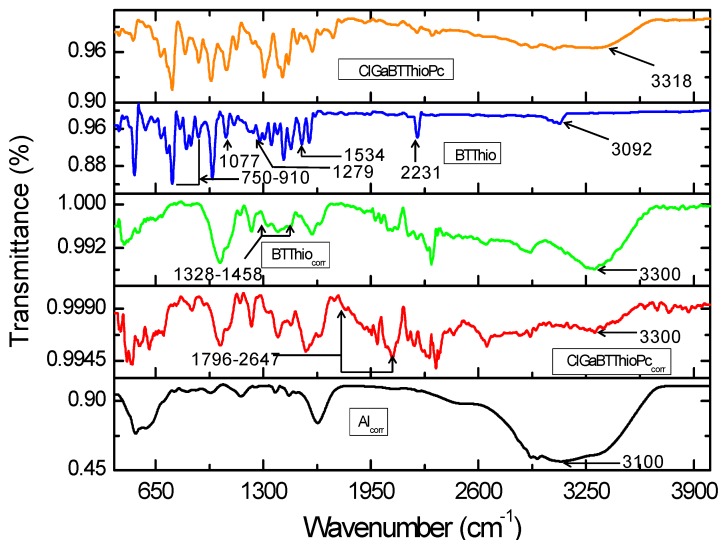
Infrared spectra of corroded aluminum in the absence of inhibitor (Al_corr_), corroded aluminum in the presence of inhibitors (BTThio_corr_ and ClGaBTThioPc_corr_), and the inhibitors (BTThio and ClGaBTThioPc).

**Figure 9 molecules-24-00207-f009:**
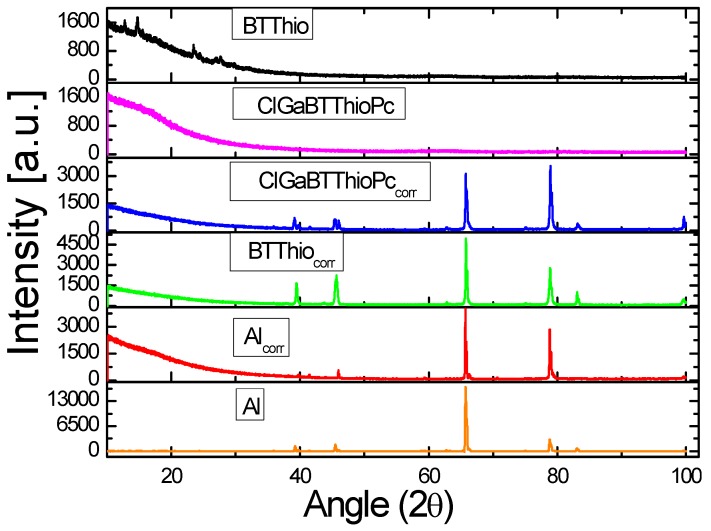
XRD patterns of: uncorroded aluminum metal (Al), aluminum metal after immersion in 1.0 M HCl (Al_corr_), aluminum metal after immersion in 1.0 M HCl containing BTThio (BTThio_corr_), aluminum metal after immersion in 1.0 M HCl containing ClGaBTThioPc (ClGaBTThioPc_corr_) and inhibitors (ClGaBTThioPc and BTThio).

**Figure 10 molecules-24-00207-f010:**
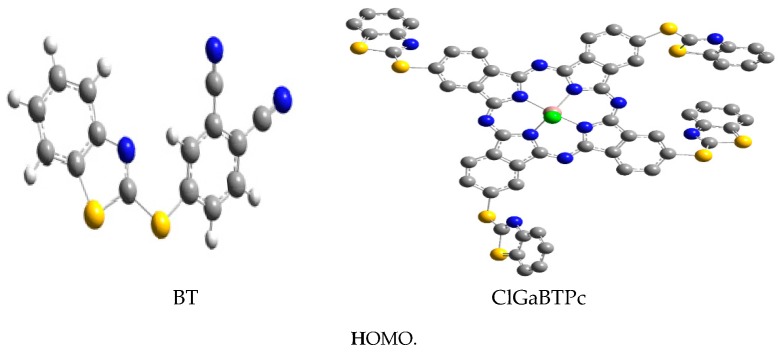
Gas phase optimised structures of studied molecules at B3LYP/6-31++G(d,p) level, highest occupied molecular orbitals (HOMO) and lowest unoccupied molecular orbitals (LUMO.

**Figure 11 molecules-24-00207-f011:**
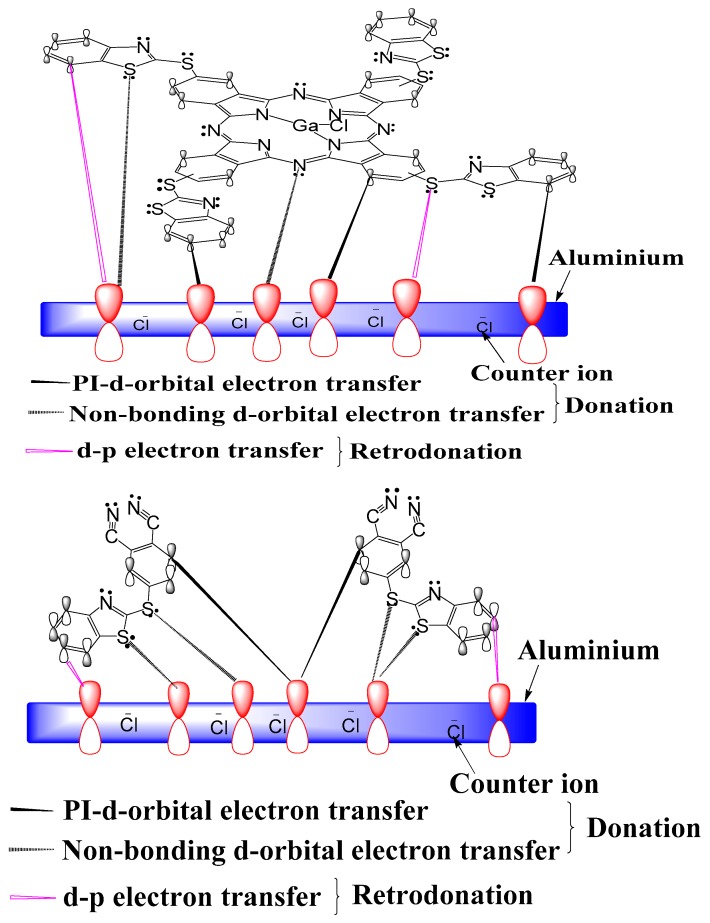
Proposed mechanism of adsorption of studied inhibitors on aluminum in hydrochloric acid solution.

**Table 1 molecules-24-00207-t001:** Polarization parameters for aluminum in 1.0 M hydrochloric acid in the absence and presence of different inhibitor concentrations.

Concentration (µM)	−Ei_corr_ (mA cm^−2^)	−*β*_a_ (mV dec^−1^)	IE%		
0	672.0 ± 4.0	8.68 ± 0.44	493.8 ± 3.1	91.8 ± 1.6	-
**BTThio**					
2	652.0 ± 8.0	4.64 ± 0.8	362.3 ± 4.8	231.1 ± 7.1	46.9 ± 6.5
4	696.0 ± 6.0	4.13 ± 0.24	369.8 ± 5.5	239.7 ±1.3	52.5 ± 0.35
6	688.0 ± 16.0	3.48 ± 0.66	201.3 ± 4.3	129.1 ± 2.8	59.9 ± 1.8
8	700.0 ± 6.0	3.04 ± 0.11	279.4 ± 3.7	219.4 ± 6.3	64.9 ± 3.8
10	676.0 ± 14.0	2.54 ± 0.33	278.1 ± 2.1	236.8 ± 5.5	70.8 ± 1.3
**ClGaBTThioPc**					
2	680.0 ± 4.0	3.50 ± 0.25	267.5 ± 3.8	131.6 ± 3.6	59.7 ± 0.8
4	692.0 ± 3.0	2.65 ± 0.23	139.5 ± 2.9	79.1 ± 2.8	69.5 ± 0.9
6	692.0 ± 4.0	2.28 ± 0.10	94.8 ± 4.3	71.7 ± 7.8	73.8 ± 0.1
8	732.0 ± 3.0	1.65 ± 0.05	112.4 ± 1.4	172.2 ± 1.4	81.0 ± 0.2
10	664.0 ± 4.0	1.72 ± 0.05	157.8 ± 4.2	69.9 ± 0.5	80.2 ± 0.6

**Table 2 molecules-24-00207-t002:** Adsorption parameters of BTThio and ClGaBTThioPc onto aluminum in HCl at room temperature 28 °C ± 0.05 °C.

Isotherm	Equilibrium Constant (K) (M^−1^) ^b^	Free Energy of Adsorption (kJmol^−1^)	Adsorption Constants	Fit to Equation	χ^2^
**BTThio**					
Langmuir	5.0 × 10^5^	−42.9	-	No	10.9779
Freundlich	13.1	−16.5	n = 3.9	Yes	0.1679
Temkin	1.07 × 10^7^	−50.6	f = 6.8	Yes	0.3001
El-Awady	2.5 × 10^3^	−29.6	Y_El_ = 0.6	Yes	0.3677
Frumkin	3.4 × 10^−7^ M ^a^	+27.2	α = 1.6	No	15.5670
**ClGaBTThioPc**					
Langmuir	1.0 × 10^6^	−44.6	-	No	5.7350
Freundlich	7.7	−15.2	n = 5.1	Yes	0.1134
Temkin	4.1 × 10^7^	−53.9	f = 7.4	Yes	0.1130
El-Awady	1.0 × 10^4^	−33.1	Y_El_ = 0.7	Yes	0.1136
Frumkin	1.5 × 10^−7^ M ^a^	+29.3	α = 2.5	No	10.0029

^a^ Different units of equilibrium constants, ^b^ K_L_ for Langmuir, K_F_ for Freundlich, K_T_ for Temkin, K_El_ for El-Wardy and K_Fr_ for Frumkin.

**Table 3 molecules-24-00207-t003:** Parameters from EIS data fitting for aluminium in 1.0 M hydrochloric acid in the absence and presence of inhibitors.

Concentration (µM)	n	R_t_ (Ωcm^2^)	IE%
0	0.939 ± 0.005	26.6 ± 0.16	-
**BTThio**			
2	0.924 ± 0.000	48.81 ± 0.65	45.5 ± 0.4
4	0.928 ± 0.100	54.22 ± 1.76	50.9 ± 1.3
6	0.927 ± 0.000	61.15 ± 0.37	58.6 ± 2.1
8	0.934 ± 0.000	72.91 ± 2.10	63.5 ± 0.8
10	0.922 ± 0.000	85.09 ± 1.83	68.7 ± 0.5
**ClGaBTThioPc**			
2	0.924 ± 0.000	59.52 ± 0.58	55.3 ± 0.7
4	0.940 ± 0.016	85.24 ± 4.39	68.7 ± 1.8
6	0.940 ± 0.000	114.92 ± 16.26	76.4 ± 3.2
8	0.901 ± 0.080	157.40 ± 0.36	83.1 ± 0.14
10	0.922 ± 0.040	140.87 ± 3.78	81.1 ± 0.62

**Table 4 molecules-24-00207-t004:** Quantum chemical parameters for ClGaBTThioPc and BTThio obtained at B3LYP/6-31++(d,p) level.

Inhibitor Molecule	E_HOMO_ (eV)	E_LUMO_ (eV)	ΔE (eV)	η (eV)	χ (eV)	ΔN	δ (eV)
ClGaBTThioPc	−6.12	−3.66	2.46	1.23	4.89	−0.83	0.81
BTThio	−8.14	−2.56	5.59	2.79	5.35	−1.06	0.358

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
