# Peer review of "Corrosion Resistance of Aluminum against Acid Activation: Impact of Benzothiazole-Substituted Gallium Phthalocyanine"

_molecules, 2019, doi:10.3390/molecules24010207_

Round 1
Reviewer 1 Report
This manuscript was carefully revised; structured and major explanations were also inserted into final manuscript. English writing was reasonably checked and improved. In my opinion the paper deserves publication in this final form.
Author Response
Comment: This manuscript was carefully revised; structured and major explanations were also inserted into final manuscript. English writing was reasonably checked and improved. In my opinion the paper deserves publication in this final form.
Response: Thank you
Reviewer 2 Report
In the revised version, the authors responded to all my comments and concerns.
My proposal will be to accept the paper for publication, in the present form.
Author Response
Comments: In the revised version, the authors responded to all my comments and concerns.
My proposal will be to accept the paper for publication, in the present form.
Response: Thank you
Reviewer 3 Report
Even authors did not care about the suggestion of earlier comments. I did not find any significant improvement in the manuscript. The reply of authors are not scientific. Thus I must reject this manuscript again.
Author Response
Comment: Even authors did not care about the suggestion of earlier comments. I did not find any significant improvement in the manuscript. The reply of authors are not scientific. Thus I must reject this manuscript again.
Response: We did seriously take all reviewers comments seriously. The original comments from Reviewer 3 were all complied with, including adding EIS data and discussing it. We have added more discussion on the novelty
Reviewer 4 Report
The authors have done a good work that could provide valuable information in this field. However, there are some issues need to be resolved.
Line 15, "character" is not properly used here, may be changed to "characteristic". Please check throughout the manuscript for the same errors and correct them all.
Line 15, please check the use of "for" in this case.
Line 16, "by" should be changed to "with" in this case. Please check throughout the manuscript for the same errors and correct them all.
Line 28, please check the use of "for" in this case.
Line 34, "it" can be deleted here.
Line 44 to 45, the sentence is not necessary here and can be deleted.
In the section of Introduction, the authors have introduced the background in this field; however, it is not very clear that the goal of this work. Please consider clarifying the goal in the end if Introduction.
"Fig. x" is used in the text, however in the captions of figures, "Figure x" is used. Please check throughout the manuscript for all the similar issues and make them all consistent in format.
In Figure 1, the two images need sub-captions and sub-captions.
Line 67, what is the shape of the exposed area? if is an orifice, then what is the diameter?
Line 72, "before" can be deleted here.
Line 76, please check the use of "coupled to".
Line 82, what does "it" mean here? please check the sentence for grammatical errors.
Line 98 to 99, the content sounds like repeated.
Line 108, please check if "corrosion current" should be "corrosion current density" here?
Line 109, "before" can be deleted here.
Equation Sx are not easy to be found. The authors should introduce where they can be found at the beginning. Equation Sx and Equation x are very confused. Please check and arrange them to be easy for the readers.
In the text, "equation x" should be written as "Equation x". Please check throughout the manuscript for the same issues and fix them all.
Line 114, please indicate what the percentage of error is acceptable and provide the supporting references.
Line 131 to 133, the sentence sounds repeated.
Line 135, add an article "the" for "influence". Please check throughout the manuscript for the same issues and fix them all.
Line 135, "to" should be changed to "on" in this case.
In Figure 2, the sub-figures are labeled "A" and "B". However, in other figures, for example, Figure 3, the sub-figures are "a" and "b". Please check throughout the manuscript and make all sub-captions in figures and text to be consistent in format.
Line 178, the last sentence sounds not necessary here.
Line 205, please check if "Fig. 5" should be "Fig. 4". If so, please check throughout the manuscript for all the related contents and correct them all.
Figure 4 needs a general caption.
The images in Figure 5 need scale bars.
Line 345, "loop" could be changed to "curve" or "circle" in this case. Please check throughout the manuscript for the same issues and fix them all.
Line 389, there are two "that" please check and fix.
Line 510, please check the use of the semicolon ";" in the sentence and fix the sentence for the grammatical issue.
Figure 10 needs sub-figures and sub-captions.
Figure 10 is provided but has been not mentioned in the text. Please check and fix.
Figure 11 needs sub-figures and sub-captions.
Line 567, should EIS be mentioned here?
Line 576, "by" can be changed to "with" in this case.
There are many grammatical errors, I have not pointed them all in this review. I suggest the authors have the manuscript have professional English proofreading.
Author Response
Comments and Suggestions for Authors
The authors have done a good work that could provide valuable information in this field. However, there are some issues need to be resolved.
Response: Thank you.
Line 15, "character" is not properly used here, may be changed to "characteristic". Please check throughout the manuscript for the same errors and correct them all.
Response: Thank you, we have rephrased the sentence
Line 15, please check the use of "for" in this case.
Response: Thank you, we have rephrased the sentence
Line 16, "by" should be changed to "with" in this case. Please check throughout the manuscript for the same errors and correct them all.
Response: Thank you, we have rephrased the sentence
Line 28, please check the use of "for" in this case.
Response: We have replaced “for” with “in”
Line 34, "it" can be deleted here.
Response: It is deleted now
Line 44 to 45, the sentence is not necessary here and can be deleted.
Response: Deleted as suggested
In the section of Introduction, the authors have introduced the background in this field; however, it is not very clear that the goal of this work. Please consider clarifying the goal in the end if Introduction.
Response: The aim and/or objective section has been included, see page 2
"Fig. x" is used in the text, however in the captions of figures, "Figure x" is used. Please check throughout the manuscript for all the similar issues and make them all consistent in format.
Response: Done, changed to Figure
In Figure 1, the two images need sub-captions and sub-captions.
Response: (a) and (b) added and described in the caption
Line 67, what is the shape of the exposed area? if is an orifice, then what is the diameter?
Response: Included
Line 72, "before" can be deleted here.
Response: Deleted
Line 76, please check the use of "coupled to".
Response: Re-worded
Line 82, what does "it" mean here? please check the sentence for grammatical errors.
Response: Now rephrased (Page 3).
Line 98 to 99, the content sounds like repeated.
Response: Re-phrased (Page 3).
Line 108, please check if "corrosion current" should be "corrosion current density" here?
Response: Corrected, to current density throughout, thank you
Line 109, "before" can be deleted here.
Response: Deleted (Page 3).
Equation Sx are not easy to be found. The authors should introduce where they can be found at the beginning. Equation Sx and Equation x are very confused. Please check and arrange them to be easy for the readers.
Response: Equations Sx are put in the supplementary information and appropriately mentioned. We have now also indicated in the supporting information which equation is for which isotherm. We have also re-introduced the equations in the text, page 6.
In the text, "equation x" should be written as "Equation x". Please check throughout the manuscript for the same issues and fix them all.
Response: Corrected.
Line 114, please indicate what the percentage of error is acceptable and provide the supporting references.
Response: Now included (Page 3).
Line 131 to 133, the sentence sounds repeated.
Response: Sentence has been removed
Line 135, add an article "the" for "influence". Please check throughout the manuscript for the same issues and fix them all.
Response: Done (Page 4).
Line 135, "to" should be changed to "on" in this case.
Response: Corrected (Page 4).
In Figure 2, the sub-figures are labeled "A" and "B". However, in other figures, for example, Figure 3, the sub-figures are "a" and "b". Please check throughout the manuscript and make all sub-captions in figures and text to be consistent in format.
Response: Corrected for all figures to (a), (b) etc. Also corrected in the captions
Line 178, the last sentence sounds not necessary here.
Response: Removed (Page 5).
Line 205, please check if "Fig. 5" should be "Fig. 4". If so, please check throughout the manuscript for all the related contents and correct them all.
Response: Corrected (Page 6).
Figure 4 needs a general caption.
Response: Corrected (Page 7).
The images in Figure 5 need scale bars.
Response: Done (Page 8).
Line 345, "loop" could be changed to "curve" or "circle" in this case. Please check throughout the manuscript for the same issues and fix them all.
Response: Corrected (Pages 11 and 12).
Line 389, there are two "that" please check and fix.
Response: Corrected (Page 12). The second “that” removed
Line 510, please check the use of the semicolon ";" in the sentence and fix the sentence for the grammatical issue.
Response: Corrected as suggested (Page 15). The sentence has been re-worded.
Figure 10 needs sub-figures and sub-captions.
Response: Done (Pages 16 and 17), (a), (b), (c) to the figure and an defined in the figure caption.
Figure 10 is provided but has been not mentioned in the text. Please check and fix.
Response: Mentioned (Page 15). Figure 10 was already cited.
Figure 11 needs sub-figures and sub-captions.
Response: Corrected (Page 18). a), (b) to the figure and an defined in the figure caption.
Line 567, should EIS be mentioned here?
Response: Included, thank you (Page 18).
Line 576, "by" can be changed to "with" in this case.
Response: Corrected as suggested (Page 19).
There are many grammatical errors, I have not pointed them all in this review. I suggest the authors have the manuscript have professional English proofreading.
Response: The manuscript has been read again and any errors found have been corrected.
Round 2
Reviewer 3 Report
now it can be accepted